# Fatty Acids and Metabolomic Composition of Follicular Fluid Collected from Environments Associated with Good and Poor Oocyte Competence in Goats

**DOI:** 10.3390/ijms23084141

**Published:** 2022-04-08

**Authors:** Dolors Izquierdo, Montserrat Roura, Míriam Pérez-Trujillo, Sandra Soto-Heras, María-Teresa Paramio

**Affiliations:** 1Departament de Ciència Animal i dels Aliments, Facultat de Veterinària, Universitat Autònoma de Barcelona, 08193 Bellaterra, Spain; dolors.izquierdo@uab.cat (D.I.); monroura@gmail.com (M.R.); 2Servei de Ressonància Magnètica Nuclear, Facultat de Ciències i Biociències, Universitat Autònoma de Barcelona, 08193 Bellaterra, Spain; miriam.perez@uab.cat; 3Department of Animal Sciences, University of Illinois Urbana-Champaign, Urbana, IL 61820, USA; sandras6@illinois.edu

**Keywords:** reproductive fluids, fertilization, follicles, omega-3, omega-6, glucose

## Abstract

In goats, embryo oocyte competence is affected by follicle size regardless the age of the females. In previous studies we have found differences in blastocyst development between oocytes coming of small (<3 mm) and large follicles (>3 mm) in prepubertal (1–2 months-old) goats. Oocyte competence and Follicular Fluid (FF) composition changes throughout follicle growth. The aim of this study was to analyze Fatty Acids (FAs) composition and metabolomic profiles of FF recovered from small and large follicles of prepubertal goats and follicles of adult goats. FAs were analyzed by chromatography and metabolites by 1H-Nuclear Magnetic Resonance (1H-NMR) Spectrometry. The results showed important differences between adult and prepubertal follicles: (a) the presence of α,β-glucose in adult and no detection in prepubertal; (b) lactate, -N-(CH3)3 groups and inositol were higher in prepubertal (c) the percentage of Linolenic Acid, Total Saturated Fatty Acids and n-3 PUFAs were higher in adults; and (d) the percentage of Linoleic Acid, total MUFAs, PUFAs, n-6 PUFAs and n-6 PUFAs: n-3 PUFAs ratio were higher in prepubertal goats. Not significant differences were found in follicle size of prepubertal goats, despite the differences in oocyte competence for in vitro embryo production.

## 1. Introduction

Oocyte competence or oocyte quality is defined as the ability to resume meiosis (or mature), to cleave following fertilization, to develop to the blastocyst stage and to give rise to normal and fertile offspring after gestation. Oocyte competence is acquired gradually during the course of folliculogenesis as the oocyte grows and its somatic cell cohort differentiates [1]. Oocyte competence play a principal role in the results of the In vitro embryo production (IVEP) and Juvenil in vitro embryo technologies (JIVET) programs. JIVET will allow to have offspring of prepubertal females. The JIVET into breeding programs could reduce the generation interval and increase the rate of genetic gain. However, the greatest limitation of JIVET is the lower production of embryos compared to embryo development of oocytes from adult females (reviewed by [2]). It is well known the positive and direct relationship between follicle size, oocyte diameter and embryo development (reviewed by [3].) In goats, the relationship between follicle size and oocyte competence to develop to the blastocyst stage was observed by Crozet et al. [4], who obtained blastocyst percentages of 6%, 12%, 26%, and 41% for oocytes from follicles of 2 to 3 mm, 3.1 to 5 mm, larger than 5 mm and ovulated oocytes, respectively.

In prepubertal goats (1 to 2 month-old) we have observed only 1.1 follicles larger than 3 mm per ovary and none larger than 6 mm [5]. Thus, our hypothesis is that oocytes from prepubertal females have a low embryo development because of the small follicles from which they originate. In a previous study in prepubertal goats, we have observed the blastocyst development of oocytes coming from follicles larger than 3 mm were significantly higher than those coming from follicles smaller than 3 mm (18 vs. 5%; respectively). In this study, we did not observe differences between blastocyst rates of oocytes coming from large follicles of prepubertal goats and oocytes recovered from ovaries of adult goats by LOPU (Laparoscopic Ovum Pick Up) (21% blastocyst) [6]. Oocytes of prepubertal females are routinely obtained by slicing the ovary and selecting the oocytes by diameter and morphology of cumulus-oocyte-complexes (COCs). Thus, Anguita et al. [7] classifying prepubertal goat oocytes diameters in 4 categories: <110 μm; 110–125 μm; 125–135 μm and larger than 135 μm observed a blastocyst development of 0%, 0%, 7% and 10%, respectively, after IVF. Using Intracytoplasmic Sperm Injection (ICSI), Jimenez-Macedo et al. [8] found significant differences between oocytes 125–135 μm diameter and larger than 135 μm in cleavage (67% and 75% respectively) but blastocyst yield was not different (16% and 11%, respectively).

Oocyte competence and Follicular Fluid (FF) composition changes throughout follicle growth and development. Follicular fluid composition is a mixture of serum exudation and substances secreted by oocyte and follicular cells. Thus, the composition of follicular fluid reflects the physiological status of follicle, and its characterization is of major interest to understand follicular development and the acquisition of oocyte competence [9].

Follicular fluid has the potential as a biomarker for oocyte quality as it reflects the oocyte in vivo microenvironment [10]. It is a well-known fact that mammalian oocytes can resume meiosis spontaneously when removed from the follicular environment, which means that the components of immature or small follicles and oocyte surrounding somatic cells are responsible for regulating oocyte meiotic arrest and that in the growing or maturing antral follicle there are the components needed for oocyte nuclear and cytoplasmic maturation. The surrounding environment and the somatic cells that accompany the oocyte are critical for oocyte health [11]. One of the FF compounds affecting oocyte competence are Fatty Acids (FA). Fatty Acids have the capacity to generate 5–6-fold more energy than glucose (reviewed by [12]), which means that this form of metabolism for generating ATP is highly efficient. Specific FAs and their concentrations inside of follicle are related with female fertility in different species including humans (reviewed by [13]).

Omics technologies (like genomics, proteomics, metabolomics) provide an opportunity to identify potential biomarkers in follicular fluid ([14]; cow, pig and horse). Metabolomics is defined as the non-targeted identification and quantification of all low molecular weight end-products of metabolism (metabolites). The High-resolution proton nuclear magnetic resonance (1H-NMR) spectroscopy is a unique tool for studying the composition of biofluids as it provides qualitative and quantitative data of all the metabolites present in a sample [9]. NMR is an ideal platform to analyze biofluids because it is the only method that requires little or no sample preparation, is nondestructive to the sample and requires small sample volumes [15]. This is a significant benefit because FF are not available in large volumes.

Within this context, the proposed study aimed to reveal whether there are changes in the follicular fluid of FA and metabolite profiles regarding follicular diameter and goat age to find the potential biomarkers associated with oocyte competence to develop up to blastocyst stage after an IVEP program.

## 2. Results

### 2.1. Fatty Acids (FA) Composition of Follicular Fluid (FF)

The Table 1 shows the percentage of the different Fatty Acids (FAs) found in the FF obtained from prepubertal and adult goat ovaries and classified in follicular fluid (FF) from small (smaller than 3 mm) and large (larger than 3 mm) follicles. At this table, we can observe that in adult goats the order of FA abundance was Palmitic; Stearic; Oleic; Linoleic and Arachidonic Acids. In prepubertal goat was Oleic; Palmitic; Stearic; Linoleic and Arachidonic Acids. In general, in adult and prepubertal goats there was no difference among the FA composition of large (≥3 mm) and small follicles (<3 mm).

Regardless of the follicular size, we observed that prepubertal females compared to adult females had statistically lower percentages of omega 3 Linolenic Acid (ALA: 1.1 vs. 2.3%, respectively), total Saturated Fatty Acids (SFA: 42.6 vs. 51.4% respectively), and omega 3 Polyunsaturated FAs (n-3 PUFAs: 3.8 vs. 6.4% respectively). Contrarily, adult females had, in comparison with prepubertal females, lower percentages of omega 6 Linoleic Acid (LA: 10.3 vs. 12.0%, respectively), total Monounsaturated FA (MUFAs: 26.1 vs. 32.4% respectively), total Polyunsaturated FAs (PUFAs: 22.5 vs. 25.0% respectively) and omega 6 PUFAs (16.1 vs. 22.7%, respectively). Thus, the n6:n3 PUFA ratio was lower in adult females compared to its prepubertal counterparts (2.6 vs. 6.3, respectively).

Table 2 shows the concentration (μM) of the different FA in FF recovered from prepubertal goat ovaries and classified in small (<3 mm) and large (≥3 mm) follicles. At this table, we observe differences in SFA (1144.74 μM vs. 993.88 μM), MUFAs (vs 829.17 μM vs. 700.32 μM), PUFAs (608.97 μM vs. 527.20 μM) and n-6 PUFAs (524.27 μM vs. 445.32 μM) in small and large follicles, respectively. These FAs were significantly higher in small than in large follicles of prepubertal goats.

### 2.2. Metabolome Composition of FF by 1H-NMR

A comparison of the FF metabolome of adult and prepubertal goats was conducted. For that, a 1H NMR-based fingerprint of each FF sample (total samples = 56) was obtained (see Section 4). Assignments of the most intense NMR signals to their corresponding metabolite were done based on comparison of the resonance frequencies (chemical shifts, δ) and line shapes (multiplicity and coupling constants) to prior reported data [16,17,18]. Figure 1a,b show the metabolic profile of two representative FF samples corresponding to adult and prepubertal individuals, respectively. A clear difference of both metabolic profiles, directly observed by visual comparison of the spectra, was the presence of α,β-glucose in the FF of adult goats and the absence (no detection) in prepubertal ones. A multivariate ordination principal component analysis (PCA), conducted on the FF fingerprint data, differed between adults FF and prepubertal FF along PC2 (Figure 2a). The heat map representation [19] of the loadings plot data is represented in Figure 2b. Colored loadings highlighted buckets associated to identified FF metabolites with a significant change in their concentration between adult and prepubertal individuals (*p*-value < 2.45 × 10^−4^, Bonferroni corrected confidence interval, Appendix A). They corresponded to lactate (4.13 and 1.33 ppm), α,β-glucose (3.73, 3.48, 5.28, 5.33, 4.68, 3.88 and 3.43 ppm), -N-(CH_3_)_3_ groups (3.23 ppm) and inositol (4.08 ppm). Lactate, trimethylamines [-N-(CH_3_)_3_ groups] and inositol showed higher concentrations in the FF samples of prepubertal goat, whereas α,β-glucose presented higher concentrations in the FF samples of adults (Appendix A).

Analogous to the previous study, a comparison of the metabolome of the FF of prepubertal goats recovered from large and small follicles was conducted. For that, a 1H NMR-based fingerprint of each sample was obtained. Figure 3a,b show the 1H NMR metabolic profile of FF representative samples of prepubertal goats with small and large follicles, respectively. A PCA conducted on the NMR fingerprint data of prepubertal samples (n = 16) differed between the FF metabolome of samples from large and small follicles. The PC1 vs. PC2 scores plot shows the separation between the two groups along PC2 (Figure 4a). Figure 4b shows a heat map representation of the loadings plot data. Loadings in red and blue correspond to buckets associated to FF metabolites with a significant change in concentration between the two groups (*p*-value < 3.60 × 10^−4^, Bonferroni corrected confidence interval, Appendix A), which were identified as inositol (3.63 and 3.53 ppm) and lysine (3.78 ppm). Both metabolites presented higher concentrations in small follicles samples than in large follicles samples (Appendix A).

## 3. Discussion

Results of this study indicated remarkable differences in Fatty Acid (FA) and metabolic profiles of follicular fluid (FF) of goats, regarding female age but not in the follicle size, in spite of both parameters were associated with embryo development competence of oocytes.

Phospholipids and triglycerides are two major classes of lipids that exist in ovarian follicles. Fatty Acid composition is one of the important components of lipid fractions of phospholipids and triglycerides. Phospholipids act as structural constituents of cellular membranes and are implicated in signal transduction as the precursor of various messenger molecules. Triglycerides form a major part of lipid droplets contributing to energy generation and membrane biogenesis. In mammal cumulus oocyte complexes (COCs), the most common sources of energy are either glucose or Fatty Acids.

In the present study, assessing FF of adult and prepubertal goats, the percentage of Linolenic Acid (ALA; n-3 PUFA), total Saturated Fatty Acids (SFAs) and total n-3 PUFAs were significantly higher in adults; and the percentage of Linoleic Acid (LA; n-6 PUFA), total MUFAs, total PUFAs, total n-6 PUFAs and n-6:n-3 PUFA ratio were higher in prepubertal goats than in adult ones. However, in prepubertal goats, comparing large (≥3 mm) and small (<3 mm) follicles the concentrations of total SFAs, MUFAs, PUFAs and n-6 PUFAs were higher in the small ones.

In goats, both adult and prepubertal, the most abundant FA were Palmitic, Oleic, Stearic, Linoleic (LA) and Arachidonic Acids, followed by Linolenic Acid (ALA). These results are similar in sheep [20]. However, in cows the most abundant FA is Linoleic Acid (LA), followed by Oleic, Stearic, Palmitic and linolenic (ALA) Acids [21,22,23,24].

Earlier studies in cattle [25] showed that Linoeic Acid (LA) represented about a third of the total FA but this LA proportion was higher in small follicles (1 to 3 mm) and ALA was higher in large follicles (>7 mm). In our study in goats, we have observed significantly higher percentage of LA and lower ALA in prepubertal goats than adult goats. However, there were not differences between follicle sizes in any female age.

In Table 2 we observed in prepubertal goats the concentration of LA ranged from 276 µM to 287 µM and ALA concentration ranged from 23 µM to 27 µM with a LA:ALA ratio of 12 and 15 for large and small follicles, respectively with not significant differences between follicle size. In adult goats, [26] found ALA concentrations of 64.6 µM and 100.6 µM, for small (<2 mm) and large (>6 mm) follicles, respectively. In studies in vitro, ALA and LA concentration added as supplements of IVM medium affect oocyte competence for embryo development. In our laboratory, IVM media supplemented with different concentrations of ALA (50, 100, and 200 μM) in lamb oocytes we did not find differences in blastocyst development (6.9%, 11.5% and 14.0%, respectively) compared to control group (12%) but, the number of blastomeres and the apoptotic cells in blastocysts were improved with ALA addition [27]. Later and with prepubertal goat oocytes, IVM medium was supplemented with both Fatty Acids, LA and ALA, concluding that the lower LA: ALA ratio improved blastocyst development. Thus, with the LA: ALA ratio of 1, the percentage of oocytes that developed until blastocyst was 22.58%, while LA: ALA ratio of 4, the percentage of blastocysts obtained was 9.6%. This low blastocyst development in ratio 4 group was caused by a high number of polyspermic zygotes, which could suggest that high LA concentration impairs oocyte membranes [28].

Regarding the female age, the proportion of n-3 PUFAs was significantly higher in adult goats, while n-6 PUFAs and n-6:n-3 PUFA ratio were significantly higher in prepubertal goats. PUFAs, including omega-6 and omega-3, are essential FAs but cannot be synthesized in the body and must be provided by diet. The diet in prepubertal goats was milk from the mother while grazing adult goats were fed indoor with alfalfa “ad libitum”. In cattle, females which have received the diet supplementation with 1% dry matter of n-3 PUFAs, the FF was 1.2-fold higher in n-3 PUFAs and their oocytes have developed higher quality blastocysts compared to control animals [29]. These authors concluded that supplementation of the diet with n-3 PUFAs led to an increase in the good blastocyst rate compared to control cows supplemented with n-6 PUFAs and even after ovarian stimulation. Moreover, the authors concluded that this supplementation could therefore be a benefit to the in vitro embryo production program. In bovine, Bender et al. [22] observed that the total PUFA fraction and the percentage contribution of PUFAs to the total Fatty Acid pool are significantly higher in follicular fluid from cows compared with heifers. n-6 PUFAs were dominant in terms of both the absolute concentration and their percentage contribution in cows, whereas the n-3 PUFAs had a higher percentage contribution to the follicular fluid from heifers. In goats, adult fertile females present lower LA, lower n-6 PUFAs and higher ALA, n-3 PUFAs, with a lower n-6:n-3 PUFA ratio.

In conclusion, good oocyte competence seems be positively related to the composition of n-3 PUFAs and the low n6:n3 PUFA ratio. This fact was also observed in our previous study with oocytes from prepubertal goats. We have observed that blastocyst production was significantly higher in winter than autumn after IVF (16.8% and 5.5%, respectively) and after Parthenogenic Activation (22.7% and 11.5%, respectively) and these results were related with a n6:n3 PUFA ratio significantly higher in autumn (11.17) than in winter (4.23) [30]. In women this high ratio was related with low oocyte competence (reviewed by [13]).

When analyzing the interaction age-follicle size, no differences in the percentages of FAs were found due to the size of the follicle in adult females. However, in prepubertal goats, the total concentrations of SFAs, MUFAs and PUFAs were significant higher in small follicles compared to the large ones. In cattle, Bertevello et al. [31] assessing lipid composition of a dominant large follicle (LF; >7 mm) and subordinated small follicles (SFs; 3–7 mm) within the same ovaries, obtained that Follicular Fluid from SF presented 15% more SFAs, 17% more MUFAs and 35% more PUFAs than that from LF. These changes in FF were likely due to the stage specific secretions from somatic follicular cells that was in line with the differences observed from FF extracellular vesicles and gene expression of candidate genes in granulosa and theca cells between LF and SF. In lactating cows, Bender et al. [22] concluded that the high concentrations of detrimental saturated fatty in cows would have a negative impact on oocyte maturation and early embryo development.

In this study with 1–2 months old suckling goats, FAs present similar profiles to other ruminant species which means that the ovary acts a nutrient selected barrier and that somatic follicle cells presents similar lipidic metabolism regardless the age.

In our metabolomic study by 1H-NMR we observed that the differences between adult vs. prepubertal were the presence of α, β-glucose in the FF of adult goats and absence (no detection) in prepubertal females. However, trimethylamine groups, lactate and inositol concentrations were significantly higher in FF from prepubertal goats than from adult ones. Comparing follicle size in prepubertal goats, we observed more concentration of Lysine and inositol in small follicles compared with the large ones.

Most mammalian cells use glucose predominantly for ATP synthesis. Surprisingly, the oocyte itself has a low capacity for glucose metabolism due to low phosphofructokinase activity. However, cumulus cells have a high phosphofructokinase activity and they convert glucose to pyruvate, lactate or NADPH, which are further transferred to oocytes (reviewed by [12]). Glucose can also be utilized by the pentose phosphate pathway (PPP) to regulate oocyte nuclear maturation and redox state and by the hexosamine biosynthesis pathway to provide substrates required for processes such as cumulus expansion and cell signaling [32]. Increased glucose concentration in follicular fluid is associated with oocytes with higher potential to develop to the blastocyst stage [33]. Moreover, the finding that glucose levels increased and lactate levels decreased as follicle size increased is consistent with previous findings in cattle [14,34], sheep, buffalo and pigs [17].

In our study, we did not detect glucose in FF of prepubertal goats neither in small follicles and large follicles (>3-mm). In our first studies, we have observed the low number of “large” follicles and the absence of follicles larger than 6 mm [5]. In the ovary of adult goat, follicles are considered large when they are larger than 6 mm in size [35,36]. Annes et al. [37] have observed that glucose concentrations in the FF, lipid content in oocyte and blastocyst production increase with the increase in follicular size in bovine. These authors discuss that lactate dehydrogenase catalyses the conversion of pyruvate to lactate, which in cows was at a reduced level in large FF compared with small FF.

Lower glycolytic activity is associated with poor oocyte developmental and increased glycolytic activity in bovine COCs has been associated with increased blastocyst development. Furthermore, the increase in glycolysis, normally seen during maturation, is delayed in oocytes from prepubertal sheep and cows. In vitro studies have demonstrated that glucose metabolism via the PPP is essential for adequate oocyte developmental competence [10]. In cattle, follicles high lactate and low glucose concentrations were a good indicator of small follicles and atretic follicles [38]. These authors explain their results suggesting that in small follicles, the vascular system is poor developed, with low supply of glucose and with a rapid metabolization to lactate by metabolic active somatic cells and, in atretic follicles, degeneration of the vascular network would impair glucose delivery and lactate removal causing glucose to disappear and lactate to remain in the FF. In our studies with goats, we can confirm that all of prepubertal goat follicles are small or atretics because puberty is reached at 6 months-old and no glucose is observed in the FF regardless the follicle size.

In goats, we have also observed that -N(CH3)3 groups and inositol were significantly higher in FF from prepubertal goats than from adult ones. Comparing follicle size in prepubertal females, we observed more concentration of Lysine and inositol in small follicles compared with the large ones. In pigs, Bertoldo et al. [17] using 1H-NMR also found differences in trimethylamine groups and inositol concentration due to follicle size, obtaining higher concentrations in the fluid of small follicles. Moreover, trimethylamines content at the healthy preovulatory follicles was significantly lower than follicles at the late dominant stage in mares [39] and the presence of trymetilamines in FF has been associated with lower-quality embryos in women [40]. The change in trimethylamines content could be related to polyamine metabolism, as they are also degradation products of cellular metabolism.

Lysine is an essential amino acid and thus substrate for follicular development. In cattle, it has been observed that lysine concentration in FF is affected by oestrus stage [41] and its concentration is higher in follicular fluid from preovulatory follicles prior to oestrus and luteinised preovulatory follicles than from newly selected dominant follicles [22]. In our study lysine concentration was higher in small prepubertal goat follicles.

With regard to inositol, a C6 sugar alcohol with 9 stereoisomers, this molecule plays a very important role in many physiological functions because it is involved in lipid synthesis, structure of cell membranes and cell growth (reviewed by [42]). It is a precursor of the inositol phospholipids which are a substrate of phosphatidylinositol 3-kinases (PI3Ks), an enzyme involved in fundamental pathways for regulation of survival and activation of primordial follicles, proliferation and differentiation of granulose cells in response to gonadotropins and resumption of meiotic process (reviewed by [43]).

Myo-inositol, the steroisomer of inositol most diffused in nature, is an organic osmolyte that regulates cellular responses to hypertonic environments and it is incorporated into cell membranes as phosphatidyl-myoinositol, the precursor of inositol triphosphate (InsP3), which acts as second messenger in the transduction of several endocrine signals, including FSH, TSH and insulin (reviewed by [44]). Thus, is an essential molecule for the proper functioning of reproductive system (reviewed by [44,45]) and it is positively correlated with the amount of estradiol in FF and also correlated with embryo quality in women [46]. However, in our study, the highest concentration of inositol was found in the small follicles of prepubertal goats suggesting that inositol is involved in the regulation of diverse cellular functions including cell proliferation [46] for follicle growth.

In small ruminants, FF has been used as supplement for IVM media (reviewed by [47]) showing that FF from larger and non-atretic follicles improved blastocyst development. In our laboratory testing FF from adult and prepubertal goats as supplements of IVM media for mice oocytes, we did not find differences in blastocyst development (47 and 41% of blastocysts; respectively) despite of different FA and metabolits profiles. Moreover, compared with ovulated mice oocytes, we observed significant variations in centrosome-spindle organization in metaphase-II oocytes in vitro matured with goat FF but fertilization rates were equivalent among all of them (range from 90 to 93%) but blastocyst rate was significantly higher in ovulated oocytes (90%) [48].

## 4. Materials and Methods

### 4.1. Chemicals and Reagents

All the chemicals were purchased from Sigma-Aldrich Chemicals Co (St. Louis, MO, USA) unless otherwise specified.

### 4.2. Ovary Collection of Prepubertal Goat and Follicular Fluid Extraction

Ovaries from prepubertal (1 to 2-months old) goats were collected from a local slaughterhouse and transported within half an hour in Phosphate Buffered Saline (PBS) at 37 °C. Once in the laboratory, the ovaries were washed three times in PBS at 37 °C and kept at the same temperature in an incubator until use. Immediately, follicular fluid was recovered from small (<3 mm) or large (≥3 mm) follicles using a 20 G needle attached to a syringe. The samples were pooled and centrifuged at 500 G during 20 min. Then, the supernatant was recovered and was kept in eppendorf at −80 °C until analysis. Samples were recovered in 8 replicates. The time between slaughter and FF recovery was less than 1 h.

### 4.3. Follicular Fluid Extraction of Adult Goats by Laparoscopy

#### 4.3.1. Animal Ethics

All procedures were approved by the Ethical Commission of Animal and Human Experimentation (Spanish Government, Authorization number CEEAH 2656) under the auspices of the Ethical Commission of the Autonomous University of Barcelona.

#### 4.3.2. Laparoscopy Procedure

Eight adult females were kept in indoor conditions and fed alfalfa ad libitum during all the experiment.

Two hormonally unstimulated Murciano-Granadina goats were used for each repetition. Females were deprived of food and water for 12 h prior to laparoscopy. They were pre-medicated with buprenorphine (Buprex, 0.01 mg kg^−1^ BW i.m.; Schering-Plough S.A., Welwyn Garden City, UK) and midazolam (Dormicum, 0.2 mg kg^−1^ body weight (BW) i.m.; Roche, Spain) 15 min before induction. General anaesthesia was induced with propofol (Lipurol 1%, 4 mg kg^−1^ BW i.v.; B-Braun, Rubi, Spain) and maintained with 2–2.5% isofluonate (Isoflo, Lab. Dr. Esteve S.A., Barcelona, Spain) in 100% oxygen. An orogastric tube was inserted to prevent regurgitation and aspiration pneumonia. Laparoscopy was performed as it was previously described by [49]. The animals were restrained in dorsal recumbence; the head lower than the body on a 40° incline. The pneumoperitoneum was produced by injecting carbon dioxide through a Verress needle. A 10 mm trocar, associated to endoscope, was inserted approximately 10 cm cranial from the udder and 10 cm of the left side from the midline. A 5 mm trocar was introduced to the opposite side of the 10 mm trocar for the placement of the non-traumatic grasping forceps that was utilized to fix the ovary. A second 5 mm trocar was located to 2–3 cm from the midline and it was used to introduce the handmade follicular puncture set. The puncture set was made using a modified cannula constituted by a 21 G butterfly needle (Venofix, B. Braunn, Spain) without “wings” and final connection mounted into an Aspic of insemination (Aspic IVM Cassou, L’Aigle France) and all the system was introduced into 1 mL pipette to give rigidity to the system. The cannula was connected to a drainage line that ended in a 15 mL collection tube. The follicles were aspirated with a controlled-vacuum pump (Aspirator 3, Labotect GmbH, Rosdorf, Germany), which maintained a vacuum pressure between 25 to 30 mmHg. Follicles within 2 to 9 mm of diameter were aspirated perpendicularly to the wall of the ovary, and divided into small follicles (<3 mm) and large follicles (≥3 mm). The cannula was then washed with 500 µL of PBS to collect all the fluid. At the end of the session, the ovaries were flushed with sterile heparinised saline solution (0.9% saline with 5 U mL^−1^ heparin) (Sodium heparin 5%, ROVI S.A, Madrid, Spain). The goats received buprenorphine (Buprex, 0.01 mg kg^−1^ BW i.m) every 8 h and meloxicam (Metacam, 0.1 to 0.2 mg kg^−1^ BW i.m.; Boehringer Ingelheim, Sant cugat del Vallès, Spain) every 24 h for the next three days. Antibiotherapy was performed with amoxicillin (Duphamox L.A., 22 mg kg^−1^ BW i.m; Fort Dodge Veterinaria, Vall de Bianya, Spain) every 48 h during 5 days.

The samples were centrifuged at 500 G during 20 min. Then, the supernatant was recovered and frozen in eppendorfs at −80 °C until analysis. Samples were recovered twice in each female.

### 4.4. Fatty Acids (FA) Analysis by Chromatography

For the FA extraction, Sukhija and Palmquist [50] protocol was used with some modifications. Briefly, 200 μL of sample (100 µL FF + 100 µL PBS) with 450 μL toluene, 50 µL of nonadecanoic acid (C:19, 0.767 mg/mL in toluene) as internal standard, and 1 mL methanolic HCl (5%) was vortexed for 60 s and then warmed in a water bath for 1 h at 70 °C. Subsequently, 500 μL toluene and 1.25 mL K_2_CO_3_ (12%) was added, vortexed for 30 s and centrifuged for 5 min at 1000 G. Finally, the supernatant (organic matter) was recovered and dried with Na_2_SO_4_. The extracted samples were maintained at −20 °C until gas chromatographic analysis, on an Agilent 6890 (Agilent technologies, Waldbronn, Germany), with a chromatographic column Agilent DB23 60 m × 0.32 mm × 0.25 µm. For each sample, 2 µL were injected using pulsed splitless mode, with oven initial and final temperatures of 140 and 250 °C in 93 min. Fatty Acids were identified and quantified comparing the retention time in the columns of the samples and commercial standards (Supelco 37 FAME Mix and Supelco cis-11-Vaccenic Methyl ester; Supelco analytical, Bellefonte, PA, USA).

### 4.5. Data Processing and Statistical Analysis for FA Composition

Follicular fluid composition data were analysed using analysis of variance (ANOVA) through general linear model, with multiple pair-wise comparisons by Tukey post-hoc test. Tests for normal distribution were applied, and data with non-normal distribution were square root arcsine transformed prior to ANOVA, but results are reported as back-transformed. Results were considered significant when *p* < 0.05. Statistical analysis was carried out with SAS/STAT^®^ software v 9.3 (for windows, SAS institute Inc., Cary, NC, USA).

### 4.6. Metabolomic Analysis by 1H-Nuclear Magnetic Resonance (1H-NMR) Spectrometry

A 1H NMR-based metabolomic study of the follicular fluid (FF) of adult and prepubertal goats and recovered from large and small follicles was carried out. Samples corresponding to the FF of adult (40 samples) and prepubertal (16 samples) goats were collected and stored at it was described above.

#### 4.6.1. Samples Preparation for 1H-NMR Spectrometry

400 μL of a D2O sodium phosphate buffer (0.2 M, pH 7.4, containing 1 mM of 3-(trimethylsilyl)-[2,2,3,3-2H4]-propionic acid sodium salt, TSP) was added to each sample. The samples were then transferred to 5 mm NMR tubes (Cortecnet, Voisins-le-Bretonneux, France).

#### 4.6.2. 1H NMR Spectroscopy

NMR data acquisition for the metabolomic study was conducted through high-resolution 1H NMR spectroscopy measurements, using a Bruker AVANCE 600 spectrometer fitted with an automatic sample changer and a multinuclear triple resonance (TBI) probe (Bruker Biospin, Rheinstetten, Germany) at a field strength of 14.1 T (600.13 MHz 1H frequency). The probe temperature was set to 298.0 K. Sample handling, automation and acquisition were controlled using TOPSPIN 3.1 software (Bruker Biospin, Rheinstetten, Germany). All samples were analysed identically. A 1D 1H NMR experiment using the pulse sequence commonly termed 1D NOESY-presat [16] applying a 1H 90° pulse and with presaturation of the water residual resonance was used. Following the introduction to the probe, the sample was allowed to equilibrate (2 min). Each spectrum was acquired into 32 k data points, over a spectral width of 16 ppm, as the sum of 128 transients and with a relaxation delay of 2 s. The resulting free induction decay (FID) was multiplied by an exponential apodization function, equivalent to 0.2 Hz line broadening, prior to the Fourier transform. After that, the spectrum was automatically phased, and baseline corrected. Then, the phase and baseline correction of all spectra was checked by visual inspection and manually corrected when necessary. Spectra were calibrated to the internal reference (TSP singlet at δH 0.00 ppm). The total experimental time of the analysis was ca. 10 min per sample.

The assignment of the 1H NMR signals was conducted by comparison of the resonance frequencies (chemical shifts, δ) and line shapes (multiplicity and coupling constants, J) of the signals to prior reported data [16,17,18].

### 4.7. Data Preprocessing and Statistical Analyses of NMR Data

The AMIX 3.9.14 software package (Bruker Biospin, Rheinstetten, Germany) was used to preprocess the 1H NMR spectra and to perform the statistical analysis of NMP data. NMR spectra were binned into 0.05-ppm-wide buckets over the region 10.0–0.5 ppm. The region 4.99–4.66 ppm was removed from the analysis to avoid effects of imperfect water suppression. Spectral intensities were scaled to the total intensity. Multivariate ordination principal component analysis (PCA) was performed to detect patterns of sample ordination in the metabolomes. PCAs analyzed the 1H NMR data using the proton chemical shifts, δ(^1^H) as response variables and the different samples as cases. Statistical significance analysis was performed on each bucket of the bucket table using the significance testing and data representation approach described by Goodpaster et al. [19] and implanted in the AMIX software.

## 5. Conclusions

In conclusion, Follicular Fluid of adult goats is characterized by the high ALA, low n6:n3 PUFA ratio and the presence of glucose. FF of prepubertal goats is defined by high lactate, higher total Fatty Acids, LA and n6:n3 PUFAs ratio. This composition seems to be affected by the stage of growth and atresia and metabolic activity of the somatic follicle cells as was described in cattle. Not significant differences were found in follicle size, despite the differences in oocyte competence for in vitro embryo production.

## Figures and Tables

**Figure 1 ijms-23-04141-f001:**
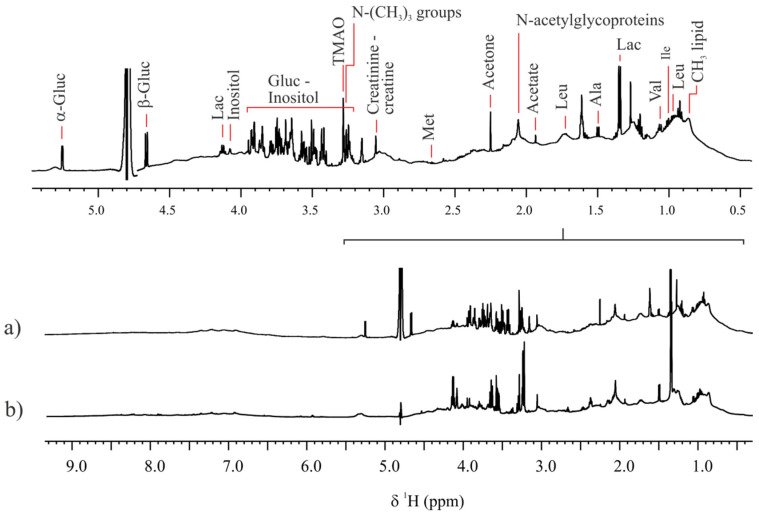
Representative 1H NMR spectra of follicular fluid samples of (**a**) adult and (**b**) prepubertal goats. Spectra were acquired at 298.0 K and at a magnetic field of 600 MHz, with suppression of the residual water signal.

**Figure 2 ijms-23-04141-f002:**
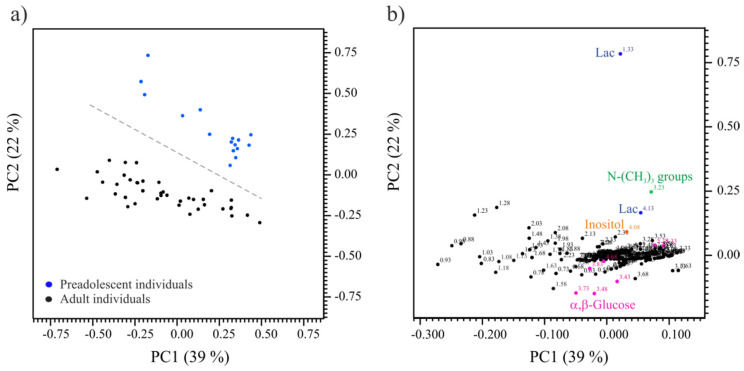
(**a**) PCA scores plot (PC1-PC2) from 1H NMR spectral data of follicular fluid samples of prepubertal (n = 16; blue dots) and adult (n = 40; black dots) goats. (**b**) PCA heat map loadings plot (PC1-PC2) with some discriminant variables assigned.

**Figure 3 ijms-23-04141-f003:**
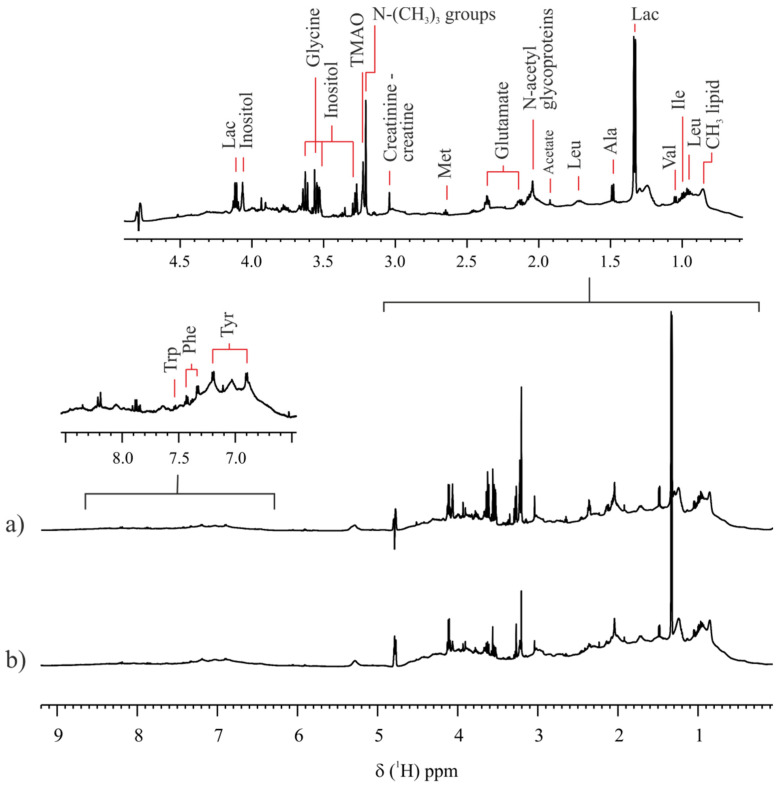
Representative 1H NMR spectra of follicular fluid samples of prepubertal goats collected from (**a**) small and (**b**) large follicles. Spectra were acquired at 298.0 K and at a magnetic field of 600 MHz, with suppression of the residual water signal.

**Figure 4 ijms-23-04141-f004:**
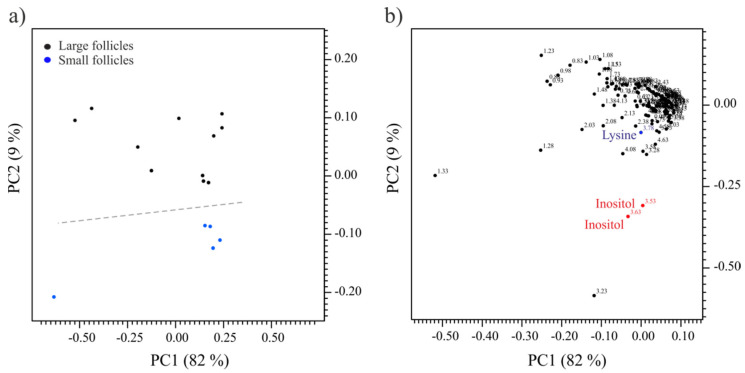
(**a**) PCA scores plot (PC1-PC2) from 1H NMR spectral data of follicular fluid samples of prepubertal goats. Blue dots correspond to samples with small follicles (n = 5) and black dots to samples with large follicles (n = 11). (**b**) PCA heat map loadings plot (PC1-PC2) with discriminant variables assigned.

**Table 1 ijms-23-04141-t001:** Percentage of Fatty Acid composition of follicular fluid according to age of the donor and follicle size.

FATTY ACID	PREPUBERTAL		ADULT	
	SMALL (<3 mm)	LARGE (≥3 mm)	SMALL (<3 mm)	LARGE (≥3 mm)
**C14:0** (**myristic**)	1.82 ± 0.08 a	1.67 ± 0.07 a	1.60 ± 0.18 ab	1.15 ± 0.17 b *
**C15:0** (**pentadecanoic**)	0.34 ± 0.03 a	0.33 ± 0.03 a	1.08 ± 0.13 b	0.75 ± 0.11 b
**C16:0** (**palmitic**)	23.88 ± 0.26	23.01 ± 0.15	24.95 ± 0.84	24.59 ± 0.62
**C16:1** (**palmitoleic**)	1.37 ± 0.06	1.41 ± 0.06	1.36 ± 0.22	1.41 ± 0.15
**C17:0** (**margaric**)	0.55 ± 0.02 a	0.64 ± 0.03 a	1.65 ± 0.11 b	1.50 ± 0.08 b
**C18:0** (**stearic**)	15.72 ± 0.27 a	17.21 ± 0.22 a	22.97 ± 0.40 b	22.66 ± 0.73 b
**C18:1n9c** (**oleic**)	27.39 ± 0.36 a	27.68 ± 0.34 a	21.35 ± 0.67 b	23.53 ± 1.00 b
**C18:1n11c** (**vaccenic**)	3.94 ± 0.15 a	3.05 ± 0.06 b	2.10 ± 0.14 c	2.47 ± 0.16 c
**C18:2n6c** (**LA**)	11.38 ± 0.51 ab	12.67 ± 0.29 a	10.60 ± 0.58 b	10.01 ± 0.45 b
**C18:3n3** (**ALA**)	0.91 ± 0.12 a	1.22 ± 0.16 a	2.32 ± 0.20 b	2.22 ± 0.16 b
**C20:4n6** (**arachidonic**)	10.04 ± 0.24 c	8.32 ± 0.23 a	5.68 ± 0.32 b	5.92 ± 0.27 b
**EPA** (**eicosapentaenoic**)	0.82 ± 0.12 a	1.02 ± 0.13 a	2.13 ± 0.27 b	1.89 ± 0.18 b
**DHA** (**docosahexaenoic**)	1.85 ± 0.18	1.78 ± 0.12	2.24 ± 0.19	1.91 ± 0.16
**SFA**	42.31 ± 0.34 a	42.86 ± 0.30 a	52.23 ± 1.19 b	50.64 ± 0.94 b
**MUFAs**	32.69 ± 0.39 a	32.14 ± 0.39 a	24.80 ± 0.76 b	27.40 ± 1.20 b
**PUFAs**	24.99 ± 0.26 a	25.01 ± 0.15 a	22.97 ± 1.15 b	21.95 ± 0.87 b
**n-3 PUFAs**	3.58 ± 0.41 a	4.02 ± 0.42 a	6.69 ± 0.62 b	6.03 ± 0.46 b
**n-6 PUFAs**	24.42 ± 0.57 a	20.99 ± 0.42 a	16.28 ± 0.73 b	15.93 ± 0.59 b
**n6:n3 PUFAs**	6.92 ± 1.22 a	5.82 ± 0.83 a	2.57 ± 0.22 b	2.75 ± 0.18 b

Values are represented as mean percentage ± SEM. Different letters in the same row (a–c) indicate significant differences (ANOVA *p* < 0.05). Letters with (*) show tendency (*p* < 0.1). SFA: Saturated Fatty Acids: myristic, pentadecanoic, palmitic, margaric and stearic acids; MUFAs: Monounsaturated Fatty Acids: palmitoleic, oleic and vaccenic acids; PUFAs: Polyunsaturated Fatty Acids: LA, ALA, EPA, DHA and arachidonic acids; n-3 PUFAs: ALA, EPA, DHA; n-6 PUFAS: LA, arachidonic acid.

**Table 2 ijms-23-04141-t002:** Fatty Acid concentration (µM) in follicular fluid of 1 to 2-month-old suckling goats, according to the size of the follicle.

FATTY ACID	FOLLICLE SIZE	
	SMALL (<3 mm)	LARGE (≥3 mm)
**C14:0** (**myristic**)	57.61 ± 4.52 a	45.21 ± 2.94 b
**C15:0** (**pentadecanoic**)	10.04 ± 1.13	8.55 ± 0.87
**C16:0** (**palmitic**)	667.80 ± 34.00 a	552.67 ± 19.74 b
**C16:1** (**palmitoleic**)	38.92 ± 3.32	34.19 ± 2.12
**C17:0** (**margaric**)	14.63 ± 1.00	14.64 ± 0.87
**C18:0** (**stearic**)	394.66 ± 14.89	372.82 ± 14.30
**C18:1n9c** (**oleic**)	694.08 ± 30.99 a	602.29 ± 17.61 b
**C18:1n11c** (**vaccenic**)	96.18 ± 6.70 a	63.83 ± 2.50 b
**C18:2n6c** (**LA**)	287.40 ± 8.91	276.87 ± 6.09
**C18:3n3** (**ALA**)	23.84 ± 3.49	27.34 ± 3.90
**C20:4n6** (**arachidonic**)	236.87 ± 13.96 a	168.44 ± 7.66 b
**EPA** (**eicosapentaenoic**)	19.86 ± 3.33	21.02 ± 2.98
**DHA** (**docosahexaenoic**)	41.00 ± 5.32	33.52 ± 2.92
**SFA**	1144.74 ± 53.66 a	993.88 ± 37.26 b
**MUFAS**	829.17 ± 39.44 a	700.32 ± 21.67 b
**PUFAS**	608.97 ± 25.78 a	527.20 ± 16.75 b
**n-3 PUFAS**	84.70 ± 11.91	81.88 ± 9.62
**n-6 PUFAS**	524.27 ± 20.05 a	445.32 ± 13.21 b
**n6:n3**	7.33 ± 1.30	6.16 ± 0.90
**LA:ALA**	15.00 ± 3.19	12.17 ± 2.13

Values are represented as mean ± SEM. Different letters in the same row (a,b) indicate significant differences (ANOVA *p* < 0.05). SFA: Saturated Fatty Acids: myristic, pentadecanoic, palmitic, margaric and stearic acids; MUFAs: Monounsaturated Fatty Acids: palmitoleic, oleic and vaccenic acids; PUFAs: Polyunsaturated Fatty Acids: LA, ALA, EPA, DHA and arachidonic acids; n-3 PUFAs: ALA, EPA, DHA; n-6 PUFAS: LA, arachidonic acid.

## Data Availability

Not applicable.

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
