# Peer review of "Fatty Acids and Metabolomic Composition of Follicular Fluid Collected from Environments Associated with Good and Poor Oocyte Competence in Goats"

_ijms, 2022, doi:10.3390/ijms23084141_

Round 1

Reviewer 1 Report

The study by Izquierdo et al. deals with the association of goat oocytes competence and follicular fluid composition regarding fatty acids content and metabolomic profile. The authors bring new information about the content of follicular fluid recovered from small and large follicles of prepubertal goats and follicles of adult goats.

I have the following comments and recommendations:

Abstract:

I recommend to the authors summarize more clearly the differences in the content of follicular fluid with respect to the age of the animal. Please consider and unify the use of capital letters in the text.  (not only in the abstract)

The last sentence in the abstract: “Not significant differences were found in follicle size, despite the differences in oocyte competence for in vitro embryo production”  contradicts the first sentence in the discussion: “Results of this study indicated remarkable differences in Fatty Acid (FA) and metabolic profiles of follicular fluid (FF) of goats, regarding follicle size and female age parameters“

Results:

Please explain the abbreviations in the legend of Tables 1 and 2: EPA, DHA

Discussion:

I recommend not starting the discussion by repeating all the results (lines 212-223), but try gradually presenting them and confronting them with the relevant literature.

Line 316-317 continue in the previous paragraph

Line 320: please explain the abbreviation PPP

Line 365 “from” instead of  “form”

Material and methods

Line 385 .... the pellet fraction was kept... this information does not make sense as the supernatant has been analyzed

How many adult animals were used in the experiment?

In lines 459-461 is stated: Samples corresponding to the FF of adult (40) and prepubertal (16) goats were collected ...

In section 4.3.2. is stated: Two hormonally unstimulated Murciano-Granadina goats were used for each repetition...... Samples were recovered in 8 replicates.

Please clarify.

Author Response

We thanks to the reviewer for his/her comments. The response of the different comments are responded in red.

The study by Izquierdo et al. deals with the association of goat oocytes competence and follicular fluid composition regarding fatty acids content and metabolomic profile. The authors bring new information about the content of follicular fluid recovered from small and large follicles of prepubertal goats and follicles of adult goats.

I have the following comments and recommendations:

Abstract:

I recommend to the authors summarize more clearly the differences in the content of follicular fluid with respect to the age of the animal. Please consider and unify the use of capital letters in the text.  (not only in the abstract). Abstract modified

The last sentence in the abstract: “Not significant differences were found in follicle size, despite the differences in oocyte competence for in vitro embryo production”  contradicts the first sentence in the discussion: “Results of this study indicated remarkable differences in Fatty Acid (FA) and metabolic profiles of follicular fluid (FF) of goats, regarding follicle size and female age parameters“

The sentence in the discussion has been changed

Results:

Please explain the abbreviations in the legend of Tables 1 and 2: EPA, DHA. Done it

Discussion:

I recommend not starting the discussion by repeating all the results (lines 212-223), but try gradually presenting them and confronting them with the relevant literature. Modified according with the suggestion

Line 316-317 continue in the previous paragraph. Modified. A sentence was missed.

Line 320: please explain the abbreviation PPP. Done it

Line 365 “from” instead of  “form” OK

Material and methods

Line 385 .... the pellet fraction was kept... this information does not make sense as the supernatant has been analyzed. Modified. Sorry, the sentence was wrong.

How many adult animals were used in the experiment?.  Added at the M&M. We have used 8 adult goats, and each one have used twice.

In lines 459-461 is stated: Samples corresponding to the FF of adult (40) and prepubertal (16) goats were collected ...Expalined

In section 4.3.2. is stated: Two hormonally unstimulated Murciano-Granadina goats were used for each repetition...... Samples were recovered in 8 replicates.

Please clarify. Explained at the M&M

Reviewer 2 Report

The presented work is devoted to the study of some aspects of the chemical composition of the follicular fluid of goats of different ages, in respect to the size of follicles. Using chromatography and 1H-nuclear magnetic resonance spectrometry, the authors found significant differences in the content of some fatty acids and low molecular weight metabolites between adult and prepubertal animals, but not between small and large follicles. These data may be useful for improving culture media for maturation of oocytes obtained from young animals.

Major comment

The follicular fluid of adult goats was collected from live animals by laparoscopy. In the case of prepubertal goats, isolated ovaries were used, which were kept for some time in an incubator. Therefore, the authors should confirm whether these two different experimental models are comparable and whether ovarian extirpation does not lead to a change in the composition of the follicular fluid. It is also necessary to report how long the isolated ovaries were kept in the incubator before sampling the follicular fluid.

Minor (technical) comments

It would be helpful if the authors provide some more details about the Juvenil in vitro embryo technologies (JIVET) program (line 36)

lines 22, 214, 330 – in the formula N-(CH3)3, the subscript must be used – N-(CH3)3

line 53 – no explanation of the abbreviation LOPU is provided

line 257 – the sentence “…to control animals and, [29]” is broken

lines 45-46 and 309-311 – the sentences duplicate each other in meaning

line 317 – the beginning of the sentence is missing

Author Response

We thanks to the reviewer for his/her comments. The response of the different comments are responded in Bold font.

The presented work is devoted to the study of some aspects of the chemical composition of the follicular fluid of goats of different ages, in respect to the size of follicles. Using chromatography and 1H-nuclear magnetic resonance spectrometry, the authors found significant differences in the content of some fatty acids and low molecular weight metabolites between adult and prepubertal animals, but not between small and large follicles. These data may be useful for improving culture media for maturation of oocytes obtained from young animals.

Major comment

The follicular fluid of adult goats was collected from live animals by laparoscopy. In the case of prepubertal goats, isolated ovaries were used, which were kept for some time in an incubator. Therefore, the authors should confirm whether these two different experimental models are comparable and whether ovarian extirpation does not lead to a change in the composition of the follicular fluid. It is also necessary to report how long the isolated ovaries were kept in the incubator before sampling the follicular fluid.

We have reported in the M& M the time lapse between slaughter and follicular fluid recovery.

In the In vitro Embryo Production programs, we used oocytes recovered from live animals and from slaughtered animals, without differences in embryo outcomes.

Minor (technical) comments

It would be helpful if the authors provide some more details about the Juvenil in vitro embryo technologies (JIVET) program (line 36). Added at the Introduction section

lines 22, 214, 330 – in the formula N-(CH3)3, the subscript must be used – N-(CH3)3. Done

line 53 – no explanation of the abbreviation LOPU is provided. Done

line 257 – the sentence “…to control animals and, [29]” is broken. Done

lines 45-46 and 309-311 – the sentences duplicate each other in meaning. Done

line 317 – the beginning of the sentence is missing. Done